# Cytotoxic and Transcriptomic Effects in Avian Hepatocytes Exposed to a Complex Mixture from Air Samples, and Their Relation to the Organic Flame Retardant Signature

**DOI:** 10.3390/toxics9120324

**Published:** 2021-11-30

**Authors:** Kelsey Ha, Pu Xia, Doug Crump, Amandeep Saini, Tom Harner, Jason O’Brien

**Affiliations:** 1Department of Chemistry and Biomolecular Sciences, University of Ottawa, Ottawa, ON K1N 6N5, Canada; kha039@uottawa.ca; 2National Wildlife Research Centre, Environment and Climate Change Canada, Ottawa, ON K1A 0H3, Canada; xiapunju@gmail.com (P.X.); Jason.Obrien@ec.gc.ca (J.O.); 3Air Quality Processes Research Section, Environment and Climate Change Canada, Toronto, ON M3H 5T4, Canada; Amandeep.Saini@ec.gc.ca (A.S.); Tom.Harner@ec.gc.ca (T.H.)

**Keywords:** organic flame retardants (OFRs), complex mixtures, in vitro screening, PCR array, passive air sampling, chicken embryonic hepatocytes (CEH)

## Abstract

Assessing complex environmental mixtures and their effects is challenging. In this study, we evaluate the utility of an avian in vitro screening approach to determine the effects of passive air sampler extracts collected from different global megacities on cytotoxicity and gene expression. Concentrations of a suite of organic flame retardants (OFRs) were quantified in extracts from a total of 19 megacities/major cities in an earlier study, and levels were highly variable across sites. Chicken embryonic hepatocytes were exposed to serial dilutions of extracts from the 19 cities for 24 h. Cell viability results indicate a high level of variability in cytotoxicity, with extracts from Toronto, Canada, having the lowest LC50 value. Partial least squares (PLS) regression analysis was used to estimate LC50 values from OFR concentrations. PLS modeling of OFRs was moderately predictive of LC50 (*p*-value = 0.0003, r^2^ = 0.66, slope = 0.76, when comparing predicted LC50 to actual values), although only after one outlier city was removed from the analysis. A chicken ToxChip PCR array, comprising 43 target genes, was used to determine effects on gene expression, and similar to results for cell viability, gene expression profiles were highly variable among the megacities. PLS modeling was used to determine if gene expression was related to the OFR profiles of the extracts. Weak relationships to the ToxChip expression profiles could be detected for only three of the 35 OFRs (indicated by regression slopes between 0.6 and 0.5 when comparing predicted to actual OFR concentrations). While this in vitro approach shows promise in terms of evaluating effects of complex mixtures, we also identified several limitations that, if addressed in future studies, might improve its performance.

## 1. Introduction

Assessment of complex mixtures that organisms are exposed to in the natural environment is a longstanding challenge [1]. Mixtures may contain tens of thousands of compounds and therefore, the analytical determination of all constituents of a mixture is not practical/feasible [2]. Thus, chemical analysis is usually limited to a restricted list of compounds of interest such as persistent organic pollutants (e.g., organic flame retardants (OFRs)) and other contaminants of emerging concern (CECs) [3,4]. However, solely measuring the concentrations of certain components within a mixture does not capture the potential cumulative biological effects associated with the mixture.

Effects-based approaches, such as in vitro bioassays, allow a direct measure of discrete endpoints following exposure to a complex mixture, which can provide insight into potential toxicity, related mechanisms of actions, and information regarding the contents of a mixture [5,6,7]. Effects-based approaches have been used in various research programs in the United States and Canada to monitor the effects of environmental mixtures on wildlife species [8,9,10,11]. For instance, Li et al. [12] integrated targeted/nontargeted chemical analysis and effects-based biomonitoring to identify priority pollutants in Great Lakes Areas of Concern using caged fathead minnows. Transcriptomic evaluation (e.g., RNA-seq and qPCR arrays) has also been used to identify potential mixture-perturbed biological pathways in wildlife and discriminate sites with high pollution from clean reference sites [11,13,14]. Effects-based assays can serve as cost-effective tools for screening complex mixtures, and specific gene expression profiles could be used to identify the presence of variable chemical classes in an ecosystem [8].

Complex mixtures of air pollutants emitted from industrial, residential and transportation sectors have raised great concerns with regard to their potential impact on human health and the environment [15,16]. Effects-based assessments have been widely used to monitor air pollution-induced biological effects [17,18]. The Global Atmospheric Passive Sampling (GAPS) network is an international program that measures contaminant concentrations and particulate matter in air samples [19]. Recently, a sub-project (GAPS-Megacities (GAPS-MC)) was initiated to monitor air contamination levels of OFRs and CECs in 20 megacities across the globe and provides a new platform to assess the global exposome in urban air [20]. In large megacities that constitute more than 50% of the world’s population, emissions of OFRs and CECs contribute significantly to the overall air pollutant burden and associated mixture toxicity [21,22,23]. To date, monitoring of persistent organic pollutants (POPs) via the GAPS-MC project has largely been based on the collection of analytical chemistry data. However, it currently lacks evaluation of the bioactivity of the complex chemical mixtures sampled from urban air from megacities.

Here, we conducted a study to evaluate whether a well-established avian hepatic in vitro screening approach represented a useful effects-based monitoring tool for assessing effects of complex mixtures. The tested mixtures were passive air sampler extracts with known concentrations of one particular class of air pollutant, OFRs, collected from 19 global megacities as part of the GAPS-MC project [20]. OFRs are widely detected in air sample mixtures, and were therefore selected as a representative chemical grouping to explore whether OFR concentrations in air samples could be correlated or estimated by effects measured using the in vitro screening approach. The objectives of this study were: (1) to determine the cytotoxic and transcriptomic effects of passive air sampler extracts from 19 global megacities (part of the GAPS-MC project) in chicken embryonic hepatocytes; and (2) to compare effects (i.e., cytotoxicity and transcriptomics profiles) with measured OFR concentrations to explore potential relationships (e.g., whether the results of the bioassay could be used to estimate chemical concentrations), which may provide novel insight into effective mixture assessment approaches.

## 2. Materials and Methods

### 2.1. Passive Air Sampler Deployment and Extract Preparation

Passive air samples were collected between March 2018 and June 2019 for two periods of 3 months each at 19 selected megacities (Figure 1; Appendix A), with populations ranging from 2–22 million [20]. The samples from period 1 were analyzed for OFRs [20], and the period 2 samples (Appendix A) were used for the current study. A previous study conducted at multiple sites in Toronto showed that there was low variability (within a factor of two) for organophosphate ester flame retardants in the polyurethane foam (PUF) samplers collected in consecutive passive sampling periods at the same sites [24]. PUF disks were deployed following previously described methods [25]. Briefly, a sampling kit including a sampling protocol, precleaned PUF disk, and a passive sampler chamber was shipped from Environment and Climate Change Canada (ECCC) to the participating partners at each of the megacity sites. Samplers were mounted, handled, and returned to ECCC for analysis by the local participants. The list of targeted analytes is available in Appendix A. Samplers were installed at a minimum of 2 m from the ground to prevent obstructed air flow and locations were far away from known OFR sources (e.g., chemical industries or highways) and in areas with limited human activity (e.g., parks). Samples were extracted using accelerated solvent extraction (ASE 350, Dionex Corporation, Sunnyvale, CA, USA) using petroleum ether and acetone solvents (83/17, *v*/*v*). Unlike the chemical analysis of OFRs in samplers from period 1, no surrogate or internal standards were added to samples for in vitro screening prior to extraction to ensure no interference with downstream toxicity testing. Each air sample extract was concentrated to 5 mL using rotary evaporation. The samples were further reduced to incipient dryness under nitrogen blow down and then solvent exchanged into 0.5 mL DMSO for in vitro screening.

### 2.2. Preparation and Dosing of Chicken Embryonic Hepatocytes

Fertilized, unincubated domestic chicken (*Gallus gallus domesticus*) eggs (*n* = 36), obtained from the Canadian Food Inspection Agency (Ottawa, ON), were incubated (Petersime Model XI) at 37 °C and 60% relative humidity until 2 days pre-hatch [26]. Embryos were euthanized by decapitation and hepatocytes were prepared by collagenase digestion and filtration of pooled liver samples, as previously described [27]. All procedures involving the handling of avian embryos were reviewed and approved by Environment and Climate Change Canada’s Wildlife Eastern Animal Care Committee (Approval Code: 21DC06; Approval Date: 13 April 2021), which is sanctioned by the Canadian Council on Animal Care (CCAC). Briefly, hepatocytes were plated in 48-well plates by adding 25 μL of the cell suspension to 500 μL of Medium 199 supplemented with sodium bicarbonate (2.24 g/L), penicillin (60 mg/L), streptomycin (100 mg/L), insulin (1 mg/L), and L-thyroxine (1 mg/L) (all reagents supplied by Sigma-Aldrich, Oakville, ON, Canada). Hepatocytes were incubated for 24 h (37 °C and 5% CO_2_) prior to dosing with 2.5 μL of the solvent control (0.5% DMSO) or serial dilutions of “neat” air extracts. The extract dilutions used for the cell viability plates were 1 (“neat”), 0.1 and 0.01. For PCR array analysis, an extract dilution of 0.1 was included, with three replicates per treatment group. Following dosing, cells were incubated for 24 h, the medium was aspirated, and the plates were either immediately frozen and stored at −80 °C for subsequent RNA isolation or assayed for cell viability. 

### 2.3. Cell Viability Determination

Cell viability was determined using the ViaLight Plus kit (Lonza Bioscience, Morrisville, NC, USA), which measures cellular ATP concentrations as a proxy for viability, according to the manufacturer’s protocol with one modification: prior to the cell lysis step, all medium was aspirated, 100 μL of fresh medium was added to each well, followed by 50 μL of lysis reagent. Viability of cells treated with air extracts was compared to untreated and DMSO controls and a positive control sample that yielded 100% cell death [26]. Luminescence was read with a 1 s integrated reading time using the Luminoskan Ascent luminometer (Thermo Fisher Scientific, Wilmington, DE, USA).

### 2.4. Chicken ToxChip PCR Array

Total RNA was extracted from CEH exposed to the highest non-cytotoxic dilution (i.e., 0.1) of the air extracts using the Qiagen RNeasy 96 kit (Qiagen, Toronto, ON, Canada) (*n* = 3 technical replicates per treatment group) following the manufacturer’s protocols, with two modifications: (a) 1 volume of 50% ethanol (instead of 70% ethanol) was used following cell lysis (to enhance RNA yield), and (b) final elution was performed with 30 μL RNAse-free water. Total RNA concentration and purity were determined by measuring absorbance at 230, 260 and 280 nm using a NanoDrop 2000 spectrophotometer (Thermo Fisher Scientific, Wilmington, DE, USA). Samples with an A260/280 > 1.8 were used for cDNA synthesis. Total RNA (~300 ng) was reverse transcribed to cDNA using the Quantitect Reverse Transcription Kit (Qiagen), according to manufacturer’s protocol, with two modifications: the 42 °C genomic DNA elimination incubation was extended to 8 min, and the 42 °C reverse transcription incubation was extended to 30 min [28]. cDNA was added to RT^2^ SYBR Green Mastermix (Qiagen) and 25 μL was aliquoted to each well of a custom-designed sixth-generation chicken ToxChip qPCR array (Qiagen Catalogue number CAPG13982). The ToxChip was built by Qiagen according to our specifications and was described in detail previously [29]. Briefly, each 96-well array contains two identical sets of 43 toxicologically relevant genes and 5 controls (Appendix A). The five controls include 2 housekeeping genes (EEF1A1 and RPL4), and 3 quality control assays (positive PCR control, genomic DNA contamination control and reverse transcription control). All arrays were run using the Stratagene MX3005P PCR system (Agilent Technologies, Mississauga, ON, Canada) as previously described [28]. No amplification was observed in the genomic DNA contamination control, and the positive PCR control met the appropriate quality control guidelines (Qiagen).

### 2.5. Data Analysis

OFR concentration data were previously reported in pg/m^3^ [20]. Here, OFR analysis was based on liquid extract concentration in pg/mL. For OFRs that had a concentration below the limit of detection, we assigned the LOD value for that chemical.

Viability of CEH dosed with air extract dilutions was compared to DMSO-treated cells. The extract dilution factors were log-transformed and the percentage viability data were fit to a nonlinear regression curve (log(dilution) vs. normalized response) using GraphPad Prism (GraphPad Prism 5.02 software, San Diego, CA, USA) to calculate LC50 values.

Raw cycle thresholds (Ct) from the Chicken ToxChip were analyzed using the 2^−ΔCT^ method [30]. Raw Ct values were normalized to an internal control gene, RPL4, because Ct values were invariable between all extract samples. Normalized data were log2-transformed prior to further statistical analysis. Differential gene expression analysis compared to the DMSO control was performed using an ANOVA followed by a false-discovery-rate (FDR) adjusted multiple comparison t-test. Genes were considered significantly dysregulated if FDR adjusted *p*-value < 0.05 and fold change > 2. Cluster analysis and principal component analysis (PCA) of OFR chemical concentrations and gene expression fold changes across the different megacities were performed using the R package, ComplexHeatmap [31].

To determine if cytotoxicity could be estimated from OFR concentrations, or if OFR concentrations could be estimated using gene expression information, partial least square regression (PLS) was performed on paired chemical and bioactivity data [11]. PLS is a multiple linear regression model associated with statistical methods including principal components regression [32]. Briefly, Y is an output object with m variables, and X is the input matrix with *p* predictor variables. PLS performs simultaneous decomposition of X and Y into latent variables (T) and associated loading vectors (Q). Regression is performed on these components, thus [32]
Y = TQ + E 
where Q is a matrix of regression coefficients (loadings) for T, and E is a matrix of error terms. For estimating bioactivity, Y was LC50, and X was OFR concentrations. For estimating OFR concentrations, Y was the concentration of OFRs, while X was the PCR array (normalized 2^−Ct^) or LC50 data. A two-step process was used for model optimization: first, a preliminary model was derived using all available predictor variables (OFRs or genes, depending on the analysis). This model was assessed for irregularities (e.g., outliers) prior to optimization. The preliminary model was optimized by keeping only the predictor variables that had variable importance in projection (VIP) values > 1. A leave-one-out cross validation was used to predict all response variables (LC50, or specific OFR concentrations, depending on the analysis). Model performance was assessed by comparing the regression slope and root mean square error (RMSE) of prediction when comparing predicted responses to actual responses.

## 3. Results and Discussion

### 3.1. Organic Flame Retardant Concentrations Vary Greatly between Cities

A previous analysis revealed that levels of OFRs detected in passive air samples varied greatly among global megacities [20]. We observed similar OFR variability when evaluating these same air extracts on a pg/mL basis (Figure 2A and Appendix A). Overall, organophosphate esters (OPEs) were the most abundant OFRs detected in air samples, which is likely associated with their increased use as PBDE alternatives (and therefore, ultimate release into the environment) [20]. With the exception of Mexico City (due to its high TBPH and TBB burdens), cities from the Group of Latin America and Caribbean (GRULAC) tended to cluster together based on hierarchical clustering and PCA. Cities in other groupings did not seem to show obvious relationships in terms of their OFR profiles. Based on the PCA, Lagos, and New York had very distinct profiles compared to other cities in the study, as indicated by their separation on principal component 1 (the *x*-axis, Figure 2B). Beijing and London had uncommonly high HBCD levels. Our next step was to determine if differences in in vitro bioactivity in exposed avian hepatocytes could be related to the variable OFR concentrations of these extracts.

It is important to note that these air sample extracts are complex mixtures, comprising many more chemicals/particulate matter than just the OFRs we measured. In particular, for air samples collected from cities, polycyclic aromatic compounds (PACs) would likely contribute to the toxicological signature. Furthermore, as recently shown by the same GAPS megacity study, oxidation of parent chemicals present in air results in a multitude of transformation products, which could increase the complexity and toxicity of these mixtures [33]. However, it is impractical to characterize all components of environmental mixtures. Therefore, one of the goals of the present study was to determine if any toxicological properties could be attributed to the OFR fraction, even with limited knowledge of the remaining components of the mixture. In the following sections, we determined the cytotoxicity and gene expression profiles of these air sampler extracts in an avian cell culture model. We then evaluated whether variable OFR concentrations were associated with any of the observed in vitro effects.

### 3.2. PUF Extracts from Megacities Have Highly Variable Cytotoxicity

Cell death represents a clear adverse outcome that can be measured immediately following exposure of hepatocytes, thus providing an initial ranking of the efficacy of the extracts to elicit a response. The rank order of PUF extracts from the 19 different megacities based on LC50 values is shown in Table 1 (raw data in Appendix A). Extracts from Toronto, Canada, Sao Paulo, Brazil, and Madrid, Spain exhibited the greatest cytotoxicity in CEH. Three sites showed no change in cell viability even up to the highest extract dilution: Buenos Aires, Argentina, New York, USA, and Istanbul, Turkey. This was surprising because New York had the 2nd highest levels of total OFRs. On the other hand, the extract from Madrid was cytotoxic but was ranked 14th based on total OFR concentrations (Table 1). The Toronto PUF extract elicited the greatest response in terms of cytotoxicity (i.e., lowest LC50) and was ranked 7th in terms of OFR concentrations. There were several cities that had similar LC50 values (e.g., New Delhi, Bogota, and Sydney) with corresponding similar OFR concentrations (Table 1). In general, the variable pattern of cytotoxicity was not consistent with OFR concentrations for PUF extracts from the megacities (R^2^ < 0.01, Table 1).

### 3.3. Cytotoxicity Could Be Estimated from OFR Concentrations Using PLS Regression

General trends in OFR concentrations (total OFR, clustering or principal component values) did not correlate with LC50 (Table 1). We used PLS modeling to determine if there were any underlying patterns in OFR concentration that might explain differences in cytotoxicity. Samples that exhibited zero cytotoxicity were omitted from the model. Our initial pre-optimization model showed that Lagos was a major outlier in terms of the relationships between OFR concentrations and LC50 (Figure 3A). Lagos, along with New York (which was not used in the PLS model due to its complete lack of cytotoxicity), was also the most divergent city in the PCA analysis. These two cities had unusually low cytotoxicity (or none in the case of New York), despite their relatively high OFR burdens. Interestingly, Lagos and New York had the highest and second highest concentrations of so-called “novel flame retardants” or NFRs, respectively (Figure 2A and Appendix A). It is unclear how this would result in lower cytotoxicity, but it may warrant further investigation in future studies. Because Lagos was such an extreme outlier in our PLS model, it was removed from further analysis (Figure 3B).

A final optimization step was performed to identify the OFRs with the greatest VIP scores, which improved the model performance. The final PLS model for estimating cytotoxicity from OFR concentrations revealed a significant linear regression relationship between measured and estimated LC50 values (*p*-value = 0.0003, r^2^ = 0.66, slope = 0.76, rmse = 0.19) (Figure 3C). The top five OFRs that contributed most to the PLS estimations included pentabromoethylbenzene (PBEB), BDE-153, tris(1,3-dichloro-2-propyl)phosphate (TDCPP), tris-(2-ethyl hexyl) phosphate (TEHP) and tri-ethyl phosphate (TEP) (Appendix A). These results demonstrate a relationship between the in vitro cytotoxicity of air sample extracts and their OFR concentrations, such that approximate estimations of overt toxicity can be made based on OFR burden alone. Further research is warranted to understand the nature of this relationship, including evaluating gene expression changes (described in the next section), to further explore the contribution of OFRs to air sample extract toxicity. As previously mentioned, there are other components (e.g., particulate matter, PACs, and transformation products of parent chemicals) in the air samples, which are almost certainly contributing to the cytotoxicity. A more comprehensive chemical characterization of the extracts would help determine if this observed relationship is likely to be causative, or merely correlative.

### 3.4. Gene Expression Results Are Variable among Megacities and Not Strongly Correlated to OFR Concentration

We next investigated whether changes in the gene expression patterns in CEH exposed to the extracts had any relationship to the OFR concentrations. A chicken ToxChip PCR array (for gene list see Appendix A) was used to determine gene expression effects in CEH at the highest noncytotoxic concentration of PUF extracts (0.1) compared with a DMSO solvent control (Appendix A). Fold changes for the complete list of gene targets included on the ToxChip are available in the Appendix A. The RNA extraction procedure failed for three megacities (Tokyo, Japan; New Delhi, India; Warsaw, Poland) and therefore, they were excluded from PCR array analysis. In addition, gene expression data from three study sites (Sao Paulo, Brazil; Bogota, Columbia; Madrid, Spain) failed our quality control criteria (highly unstable housekeeping gene expression), and thus, they were also excluded. Therefore, 13 cities (from the original 19) were included for gene expression analysis.

Heatmap clustering of gene expression data showed great variability between megacity extracts (Figure 4A). In terms of broad global regions, cities from AS (Asia-Pacific) and AF (Africa) had a greater proportion of upregulated genes, whereas the opposite was true (i.e., greater downregulation) for WEOG (Western Europe and other state group). Four main clusters of megacities were evident (Figure 4A). City cluster 1 (Kolkata, Lagos, Cairo, and Santiago) comprises genes with similar expression profiles, with a greater proportion of upregulated genes compared to city cluster 3 (Beijing, Mexico City, Sydney, London, and Istanbul), which had a greater proportion of downregulated genes. City cluster 2 (Bangkok, Buenos Aires and New York) includes expression profiles similar to the DMSO solvent-control (i.e., minimal transcriptomic modulations). This was surprising, particularly for New York, because extracts from this megacity had the second highest concentration of OFRs, although the New York sample also did not induce any cytotoxicity. Finally, city cluster 4 comprises a single megacity, isolated based on its unique gene signature from all other megacities; Toronto, Canada (Figure 4A). To further characterize the overall gene expression signatures, a principal component analysis (PCA) was performed (Figure 4B). Overall, we did not observe any clear relationships between gene expression patterns and OFR burdens. Gene expression heatmap and PCA clusters did not appear to be related to the concentration of OFRs in the air extract mixtures (Table 1).

Despite the lack of overall patterns between gene expression and OFR concentrations, we did observe some expected and interesting responses. The most highly dysregulated transcripts from the ToxChip were *Alas1, Apob, Cyp1a4, G6pc, Il16, Lss, Scd, Thrsp, Tp63, Txn*, and *Ugt1a9* (Figure 4A; Appendix A). Several of these genes are associated with the xenobiotic metabolism pathway (e.g., *Cyp1a4, Alas1, Ugt1a9, Sult1b1*), which is not surprising given that these transcripts are commonly dysregulated in response to xenobiotics [34,35]. These transcripts have also been shown to be dysregulated by various complex environmental mixtures [9,36]. The upregulation of *Cyp1a4* following activation of the aryl hydrocarbon receptor (AHR) by diverse agonists has been studied extensively, as well as *Alas1,* which is induced by xenobiotics to ensure sufficient heme production for cytochrome P450 synthesis and other hemoproteins [37]. In a similar study, CEH were exposed to herring gull egg extracts and mRNA expression levels of *Cyp1a4* and *Alas1* were significantly upregulated [8], consistent with our findings for the air extract samples. A concordant upregulation was reported in CEH following exposure to extracts from passive samplers deployed in wetlands within the Athabasca oil sands region that had variable PAC concentrations [10]. Contaminants that were not measured in this study may also have contributed to the upregulation of *Cyp1a4* and therefore require further evaluation in future studies. A target gene associated with the thyroid hormone pathway, *Thrsp,* which encodes the thyroid hormone-inducible hepatic protein and is expressed in liver and adipose tissue, where it plays a role in lipogenesis regulation [38], was downregulated by various air extracts. Flame retardants, such as PBDEs, are known to disrupt this critical endocrine pathway [39]. Extracts of petroleum coke (‘petcoke’) from the Athabasca oil sands region elicited a concordant downregulation of *Thrsp* in CEH following exposure [40]. If a dysregulation in these pathways also occurred in vivo, it could lead to adverse outcomes, especially in birds, as seen with previous studies [41].

We then used PLS modeling again to determine if any relationships between gene expression bioactivity and OFR concentrations could be identified. When we included all available expression data in the analysis, no obvious relationships were observed (Appendix A). However, when we focused on the same samples that were used for the LC50-based PLS analysis, where we did observe a relationship between OFRs and bioactivity, we identified some relationships, albeit relatively weak ones. Only three of the 35 OFRs, TEHP, tri-n-butyl phosphate (TnBP) and m-TPP had a model performance regression with a slope above 0.5 but below 0.6 (Figure 5), indicating that the gene expression information contained only fairly weak predictive information about the concentrations of the OFRs in the mixtures. The top ten contributing genes for this PLS modeling were *Ca3b*, *Cdkn1a*, *Nos2*, *Batf3*, *Mat1a*, *Tp63*, *Aldh1a1*, *Mgmt*, *Msh2* and *Polk*. This indicates that the expression of the ToxChip gene set is generally not sufficient to make confident predictions about the OFR content of these complex mixtures. Given that these samples did exhibit a relationship between OFR and LC50, it is possible that gene expression relationships might be identifiable if a larger gene set and sample size is used in future studies.

### 3.5. Limitations of the Study

This study represents a first attempt to evaluate the ability and performance of an avian in vitro screening approach to assess effects (cytotoxicity and gene expression) of exposure to complex air extracts and to explore relationships between biological and chemical profiles for a particular class of chemicals, OFRs. We think this proof of concept of the general approach helps to advance the science. However, we acknowledge that there are some limitations of the study, which preclude a more in-depth evaluation of the in vitro indicators and the complex chemical mixture in the air samples, and these likely impacted the strength of relationship observed between the various endpoints evaluated (Table 1; Figure 3 and Figure 5). We think these limitations can be better addressed in future work, but were important to list here: (i)Although the toxicological analysis conducted in this study is based on a unique data set representing average air concentrations from 19 megacities from around the world, information on the contaminant content of these samples is limited. The samples have so far been analyzed for one general class of contaminants—organic flame retardants—of which organophosphate flame retardants (OPFRs) were by far the dominant class. A future priority is to analyze the samples for a broader suite of contaminants, including for instance, polycyclic aromatic compounds, which could be especially relevant to urban population exposures and related biological responses. Transformation products of commercial chemicals such as the OPFRs that are abundant in air and formed through oxidation reactions have recently been shown to significantly contribute to exposure and risk assessment [33]. This highlights challenges in associating biological responses to individual classes of “known” chemicals when extremely large numbers of “unknown” chemicals are also present in air at toxicologically relevant concentrations. A more holistic approach to contaminant mixture assessment may be needed.(ii)Although best efforts were made to collect samples that were “representative” of a large area from each city, it is likely that megacities have a degree of heterogeneity in terms of contaminant mixtures in air, depending on the location and proximity to sources. This introduces some uncertainty in the characterization and comparison of results among cities. Additionally, in the current study, contaminant analysis and toxicological assessment were conducted on consecutive three-month samples from the same site, which may have introduced additional uncertainty associated with temporal variability of contaminants in air. Future work should evaluate the degree of heterogeneity in biological response and chemical profiles for air collected in different parts of a city with differing land-use characteristics (e.g., residential, traffic, industry, commercial etc.) and at different seasons. Some work on this topic has recently begun in Toronto [24,42], but not for the other megacities investigated here.(iii)In this study, in vitro effects were determined using avian hepatic cells to provide a theoretical basis for the approach. Future work should explore other in vitro models (e.g., human/mammalian lung cells) which could be more relevant for assessing human health risks. Although liver is a commonly used organ for toxicological studies, performing this assay with a broader range of cell types (e.g., lung epithelial, cardiomyocytes) would be warranted, especially given the negative health effects of air pollution associated with cardiovascular/respiratory endpoints [43,44].(iv)The dose range used for cell viability and gene expression evaluation was limited (i.e., selected to provide an initial qualitative comparison across megacities as opposed to permitting full transcriptomics dose-response analysis or comparison of relative potencies). For example, Toronto had the lowest LC50 value (Table 1) and the most pronounced gene dysregulation (Cluster 4; Figure 3A); however, the concentration used for gene expression analysis (0.1) was similar to the median lethal concentration (0.12; Table 1).(v)Finally, gene expression data are based on a reduced transcriptome (43 genes), meaning that a large percentage of biological pathways/processes were not considered in the analysis.

Overall, we evaluated the utility of an avian in vitro screening approach to characterize the bioactivity of complex mixtures derived from air sampler extracts collected from diverse regions across the globe. Cytotoxicity results and transcriptomic profiles showed high variability between cities. OFR concentrations provided some ability to predict mixture LC50 values. However, we did not identify any convincing relationships between OFR burdens and gene expression. There were some consistent gene expression responses observed among broad geographic regions. Follow-up studies are warranted to expand the chemical space and include additional dose groups for quantitative dose-response evaluations of cell viability and gene expression. However, from these initial results, the avian in vitro screening approach shows potential to be a useful effects-based monitoring tool for evaluating effects of complex mixtures.

## Figures and Tables

**Figure 1 toxics-09-00324-f001:**
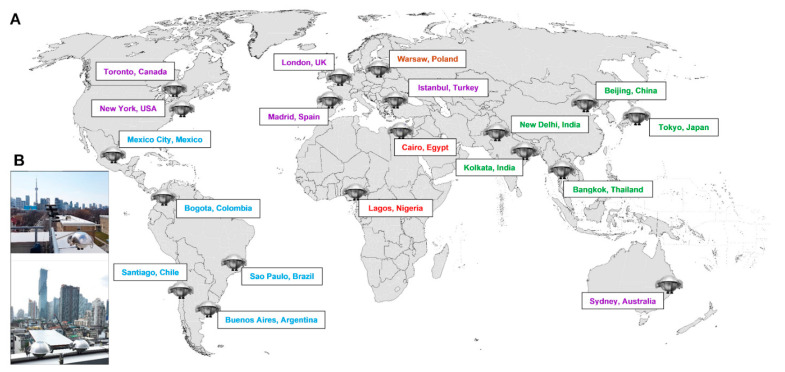
(**A**) Locations of polyurethane foam (PUF) passive air samplers for the GAPS-Megacities project (*n* = 19). (**B**) A representative example of passive air sampler deployment. Location names are coloured based on the United Nations regional groups: Africa (red); Asia-Pacific (green); WEOG = Western Europe and Other states Group (purple); GRULAC = Group of Latin American and the Caribbean (blue); CEE = Central and Eastern Europe (orange).

**Figure 2 toxics-09-00324-f002:**
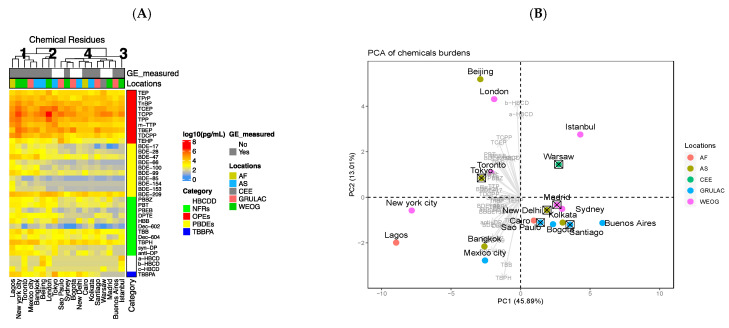
Total concentrations (log_10_ [pg/mL]) of organic flame retardants (OFRs), including organophosphate esters (OPEs), polybrominated diphenyl ethers (PBDEs), novel flame retardants (NFRs), hexabromocyclododecanes (HBCDDs) and tetrabromobisphenol A (TBBPA), measured in PUF air samplers from 19 megacities of five global regions (details in Figure 1) depicted by (**A**) hierarchical clustering, where red and blue hues indicate high and low OFR concentrations, respectively, and the grey boxes identify sites for which gene expression data were collected; and (**B**) principal component analysis (PCA) of OFR loadings, where the most influential OFRs are depicted by arrows on the graph.

**Figure 3 toxics-09-00324-f003:**
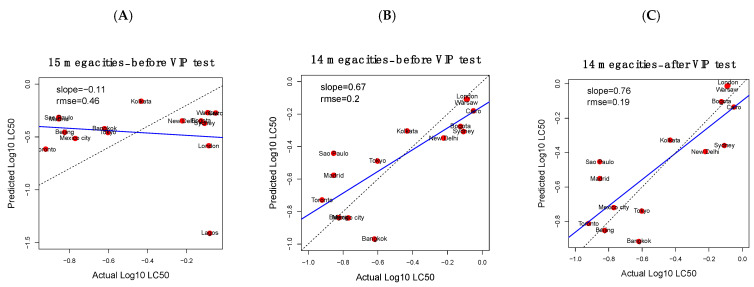
Partial least square (PLS) regression modeling for estimating cytotoxicity from organic flame retardant concentrations. (**A**) Preliminary modeling showed that Lagos was an outlier, so it was removed in the following analysis; (**B**) the preliminary model after Lagos was removed; and (**C**) the final PLS model after filtering for predictor variables with VIP scores > 1. The blue line indicates the regression between estimated and actual LC50 values. The dashed line represents a perfect 1-to-1 prediction. VIP = variable importance in projection.

**Figure 4 toxics-09-00324-f004:**
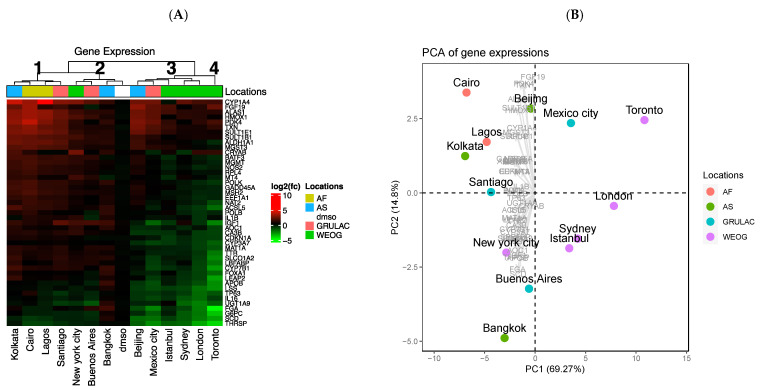
Gene expression analysis by chicken ToxChip PCR array of 13 megacities in five global regions (location details in Figure 1) depicted by (**A**) hierarchical clustering (red and green hues indicate upregulation and downregulation); and (**B**) principal component analysis. Gene expression loadings for the most responsive transcripts are depicted by arrows on the graph.

**Figure 5 toxics-09-00324-f005:**
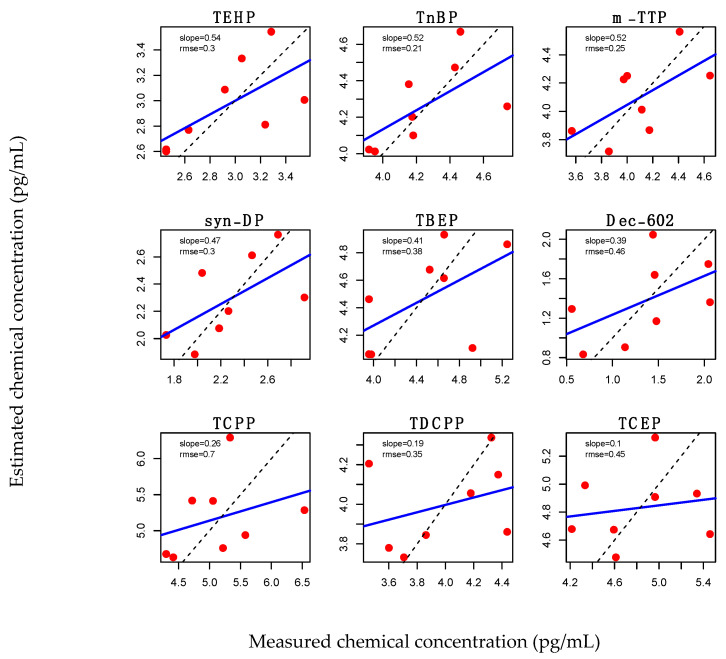
Linear regression plots comparing measured and estimated concentrations of the top nine performing OFRs identified by PLS regression when cities that did not induce cytotoxicity, and one outlier city were removed. The blue line indicates the regression between estimated and actual OFR concentrations values. The dashed line represents a perfect 1-to-1 prediction. Red dots represent the samples from each city.

**Table 1 toxics-09-00324-t001:** LC50 values for 19 global megacities. Concentration data were log-transformed and percent viability data were fit to a nonlinear regression curve to determine the LC50. ND, not determined; -, no change. Simple regression analysis was performed between all endpoints (detailed in Appendix A).

			Chemical Profile		Gene Expression Profile	
Location Name	Total OFR	LC50	Cluster ^a^	PC1 ^b^	Cluster ^c^	PC1 ^d^
Toronto, Canada	652,000	0.12	1	−2.1	4	10.83
Madrid, Spain	298,000	0.14	4	2.61	ND	ND
Sao Paulo, Brazil	682,000	0.14	4	1.43	ND	ND
Beijing, China	878,000	0.15	2	−2.9	3	−0.42
Mexico City, Mexico	304,000	0.17	1	−2.56	3	3.55
Bangkok, Thailand	458,000	0.24	1	−2.61	2	−3.03
Tokyo, Japan	1,106,000	0.25	2	−2.81	ND	ND
Kolkata, India	85,000	0.37	4	3.02	1	−6.93
New Delhi, India	150,000	0.60	4	1.89	ND	ND
Bogota, Colombia	473,000	0.76	4	3.55	ND	ND
Sydney, Australia	303,000	0.78	4	2.98	3	4.22
Warsaw, Poland	388,000	0.81	4	2.74	ND	ND
London, UK	4,606,000	0.81	2	−1.9	3	7.81
Lagos, Nigeria	747,000	0.83	1	−8.95	1	−4.80
Cairo, Egypt	188,000	0.89	4	0.95	1	−6.79
Buenos Aires, Argentina	89,000	-	4	5.89	2	−0.60
Istanbul, Turkey	250,000	-	3	4.29	3	3.38
New York, USA	3,292,000	-	1	−7.83	2	−2.85
Santiago, Chile	323,000	ND	4	2.33	1	−4.37
Regression with log (OFR)	-	*p* = 0.8628	*p* = 0.0071 *	*p* = 0.0012 *	*p* = 0.2785	*p* = 0.3131
	-	r^2^ = 0.0024	r^2^ = 0.3546	r^2^ = 0.4698	r^2^ = 0.1057	r^2^ = 0.0922
Regression with log (LC50)	*p* = 0.8628	-	*p* = 0.21893	*p* = 0.6370	*p* = 0.1235	*p* = 0.5471
	r^2^ = 0.0024	-	r^2^ = 0.1138	r^2^ = 0.0176	r^2^ = 0.3045	r^2^ = 0.0541

^a^ = based on Figure 2A, ^b^ = based on Figure 2B, ^c^ = based on Figure 4A, ^d^ = based on Figure 4B; * *p* < 0.01, ND = not determined. “-” = no effect on cytotoxicity.

## Data Availability

Raw data and all associated metadata associated with this research publication are available from Pu Xia (xiapunju@gmail.com) and Doug Crump (doug.crump@ec.gc.ca).

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
