# Peer review of "Cytotoxic and Transcriptomic Effects in Avian Hepatocytes Exposed to a Complex Mixture from Air Samples, and Their Relation to the Organic Flame Retardant Signature"

_toxics, 2021, doi:10.3390/toxics9120324_

Round 1
Reviewer 1 Report
The study under consideration has investigated the cytotoxic and transcriptomic effects of organic extracts of particulate matter deposited in passive air samplers placed in various mega cities throughout the world. Their data shows that there is great variability regarding cytoxicity and transcriptomic profiling of select genes effected by exposure of chicken embryonic hepatocytes to extracts. The effects could not be correlated with levels of a certain class of chemicals such as organic flame retardants. In summary, the study has many weaknesses (see below) and the conclusions drawn are inconclusive.
Comments
- Lung epithelial cells or lung epithelial cell lines would have been a better (more relevant) in vitro model to assess the effects of the extracts. The authors discuss this concern in the general discussion section.
- How many different samples per city were tested for their effects? A single sample per city certainly cannot be a representative sample.
- How were the extracts normalized? Per weight of sample extracted? If the samples were not normalized, how can they be compared?
- After extraction, the samples are air-dried and dissolved in DMSO. Is it assumed that all the components extracted are soluble in DMSO? What if some components are not soluble in DMSO?
- Sentence (251-253) on OFR role in cytotoxicity is contradictory.
- Gene expression results (dysregulated transcripts) were not validated by qRT-PCR and/or western blotting. How many independent samples were analyzed for changes in gene expression?
Reviewer 2 Report
The manuscript toxics-1403287 entitled “Cytotoxic and transcriptomic effects in avian hepatocytes exposed to a complex mixture from air samples, and their relation to the organic flame retardant signature” deals with the toxicity assessment of mixtures present in several highly populated cities using chicken hepatocytes as a model.
The manuscript is within the scope of the journal, however the manuscript has several issues that need to be addressed.
Can you relate your work to 3R’s concept? The use of chicken primary cells needs thorough justification. Since the whole embryos were not exposed, they only used as a cells source, why weren’t commercially available cells used? Moreover, in that case, human cells could have been used to reduce the uncertainties in translating results gained on animal cells toward humans. Additionally, why were hepatocytes used, since primary target for air pollution would be epithelial cells of respiratory system? The Ethics committee approval for the study is also missing.
Each air sample extract was concentrated (line 94), how does these concentrations reflect actual concentrations in those cities? How was 24 h exposure period chosen?
The whole idea of identifying one class of pollutants for making firm conclusions for the complex mixture is simply not convincing. Many exceptions were noted throughout the manuscript and highlighted in the limitations part. Moreover, the Lagos case in PLC model for cytotoxicity is a clear sign that other compounds drive cellular effects, and that application of such model is not suitable at this point.
The concept seems promising and prospective but, as currently written, the manuscript’s aims, results, and conclusions are in a discrepancy.
Reviewer 3 Report
Comments on toxics-1403287, “Cytotoxic and transcriptomic effects in avian hepatocytes exposed to a complex mixture from air samples, and their relation to the organic flame retardant signature”
This study looked at the toxicity of passive air samples collected in megacities from around the globe and the relation of the toxicity to organic flame retardants (OFR). This study presents an exciting concept of directly measuring the toxicity of complex mixtures in the air by running assays directly on the extracts. This is sorely needed in the current age where the complex effects of mixtures have been increasingly important for exposure assessments.
Unfortunately, I feel this paper suffers from a lack of a clear purpose. If the paper truly wanted to be about the mixture effects, it should have focused on that and not distracted the reader with OFR data. If the paper wanted to be about the effects of OFR in a mixture, it should have focused on that. I think there is some questionable QC around the OFR data in the paper (highlighted below), that really handicaps any conclusions taken from it. Ultimately, its impossible to say what the true effects of the OFR mixture was in this samples. Did the authors consider running artificial mixtures of OFR that were produced from standards at similarly ratios to what was found environmentally, to try and isolate the effects of the OFR mixtures in these assays? It would have provided valuable data about the contributions of OFRs to the mixture’s effects.
Major Concern 1: Why was OFR results included in this manuscript?
It is unclear why the organic flame retardants (OFR) data was included in this manuscript. The manuscript seems to want to talk about complex mixtures exposures with OFR specific analysis thrown in. The only justification given was on lines 58-64 which seems to indicate that it was convenience (i.e. the samples were already being analyzed for them). The authors go to great lengths to justify the OFR inclusion by multiple times reminding the reader that the OFRs are only a part of the very complex mixture (Line 196-208; Lines 248-251; Lines 326-327).
In addition to the above, the samples were not directly analyzed for OFRs which I see as a major shortcoming (Lines 78-81). Has the variability at these sampling sites been assessed? While some studies have found minimal variability across seasons for POPs, concentrations are generally seasonally dependent. This is very important implication for this study because you are trying to correlate concentrations and activity for two different sampling periods. This is only briefly addressed in the manuscript and the only justification used is a 2019 study conducted in the Greater Toronto Area. This is insufficient for the assumption that the two different sampling periods examined in this study should be the equal. This is furthermore important because you remove Lagos data as on “outlier” but your quality control is insufficient to determine whether this datapoint should be removed or kept.
I think the conclusion on lines 251-252 that cytotoxicity of air samples is correlated to OFR concentrations is overstated. First, you removed an outlier without proper justification. Why was the Lagos data point removed? Clearly it is different, but it is real data. You did not provide any justification for removing it other than the fact it makes your correlation better. Especially given that you did not directly measure concentrations in the samples but are relying on measurement from a previous period, it is hard to agree that you can rightfully remove this data point. Secondly, in general higher concentrations of compounds usually correlate to higher total mixture concentrations, so OFR concentrations may not actually be driving cytotoxicity but are just co-exposing with the active agent. To me it feels that this data was massaged (specific compounds, outliers removed) to give the desired outcome. This conclusion is even more surprising given that in the previous paragraph talk about the lack of correlation between OFR concentrations and PUF cytotoxicity, specifically saying the “In general, the variable pattern of cytotoxicity was not consistent with OFR concentrations for PUF extracts from the megacities (R2<0.01, Table 1)” (Lines 222-224).
Minor Concerns:
- Title of manuscript and SI do not match.
- Line 75: You should at least list the targeted analytes from the previous study in the SI, so all the pertinent information is contained in this manuscript.
- Line 153: Liquid extract concentrations don't really mean anything for air samples. Is the activity occurring at toxicological relevant levels in the environment? (i.e. how does it correlate to airborne concentrations?)
- Line 272: The results are underpowered if roughly 1/3 of the small sample size (6 of 19) were removed for one reason or another. Was there no way to reanalyze and recover more data.
- Line 326: This is an incorrect statement. You did not measure POPs in the extracts.
Round 2
Reviewer 1 Report
I think the authors' have responded adequately to my comments and have revised the manuscript accordingly. I recommend acceptance of the manuscript.Reviewer 2 Report
The corrections and additions made within the revision process led to significant improvement of the manuscript.
Some of the answers were incorporated to the text, so it is much more clear. The addition of limitations of the study part is valuable for readers.
I have no more comments, and would recommend acceptance.
Reviewer 3 Report
I still have concerns about the exclusion of specific data points given the analyte concentrations of the actual samples are unknown and the authors instead rely on consistent temporal trends at sampling sites. This could justification for both including and excluding the data, but rather I think it highlights a weakness in the study design.
With that being said, I think the authors adequately addressed the comments from reviewers. Adding a clear objectives section and a limitations section help provide much needed clarity in the main goals and meaning of the findings.